# The Feasibility of Interventional Pulmonology Methods for Detecting the T790M Mutation after the First or Second-Generation EGFR-TKI Resistance of Non-Small Cell Lung Cancer

**DOI:** 10.3390/diagnostics13010129

**Published:** 2022-12-30

**Authors:** Wen-Chien Cheng, Yi-Cheng Shen, Chieh-Lung Chen, Wei-Chih Liao, Hung-Jen Chen, Te-Chun Hsia, Chia-Hung Chen, Chih-Yen Tu

**Affiliations:** 1Division of Pulmonary and Critical Care, Department of Internal Medicine, China Medical University Hospital, Taichung 404, Taiwan; 2School of Medicine, College of Medicine, China Medical University, Taichung 402, Taiwan; 3Department of Life Science, National Chung Hsing University, Taichung 402, Taiwan; 4Ph.D. Program in Translational Medicine, National Chung Hsing University, Taichung 402, Taiwan; 5Rong Hsing Research Center for Translational Medicine, National Chung Hsing University, Taichung 402, Taiwan

**Keywords:** non-small cell lung cancer (NSCLC), epidermal growth factor receptor (EGFR), interventional pulmonology, T790M mutation

## Abstract

The development of third-generation epidermal growth factor receptor (EGFR)-tyrosine kinase inhibitors (TKIs) targeting T790M-mutant non-small cell lung cancer (NSCLC) has raised the importance of re-biopsy after EGFR-TKI failure. This study aimed to investigate the feasibility of interventional pulmonology (IP) procedures as re-biopsy methods for identifying the T790M mutation in EGFR-TKI-resistant patients. One hundred and thirty-nine NSCLC patients who underwent IP procedures for re-biopsy as their initial investigation after EGFR-TKI treatment failure were enrolled in this study between January 2020 and August 2022. All patients underwent a first re-biopsy with IP methods, with a diagnostic yield of 81.2% and T790M mutation detection rate of 36%. Thirty patients underwent a second re-biopsy; IP methods were used for 17 (56.6%) patients and non-IP methods for 13 (43.4%) patients; the T790M mutation detection rate was 36.4%. Only six patients underwent a third re-biopsy; no T790M mutation was noted. The T790M mutation detection rate did not differ between IP and non-IP methods (33.6 % vs. 37.5%, *p* = 0.762). In 11 cases (7.5%), a re-biopsy revealed histologic transformation from lung adenocarcinoma. IP procedures, as first-line re-biopsy methods for NSCLC, are feasible and provide sufficient tissue for identification of the resistance mechanism and target gene T790M mutation.

## 1. Introduction

The development of first-, second-, and third-generation epidermal growth factor receptor–tyrosine kinase inhibitors (EGFR-TKIs) has significantly improved the clinical outcomes of advanced EGFR-mutated non-small-cell lung cancer (NSCLC) patients [1,2,3,4,5]. The T790M mutation is the most common acquired resistance mechanism that physicians look for in EGFR-mutated NSCLC after treatment with first- and second-generation EGFR-TKIs [6]. Osimertinib, a third-generation EGFR-TKI, is effective against NSCLCs harboring the T790M mutation and is widely used in this clinical setting [7]. Osimertinib, was the preferred medicine used as first-line therapy compared to first-generation EGFR-TKI [5]. However, the price of osimertinib was high and it was not covered by health insurance in every country. The determination of a suitable standard therapy after acquired resistance to osimertinib remains a challenge. From subgroup analysis in the FLAURA study, Asian patients with L858R mutation had no significant OS benefits from osimertinib compared to first-generation EGFR TKI [5]. Sequential afatinib and osimertinib provide good outcomes in real-world clinical practice, especially in Asian patients [8,9]. In addition, Re-biopsy has a significant impact on overall survival because patients are more likely to receive third-generation EGFR-TKIs [10]. First- or second- generation EGFR-TKI was still the first-line choice for treatment of NSCLC patients with EGFR mutation and re-biopsy remained important in detecting the T790M mutation after the failure of first- and second-generation EGFR-TKI treatment in Asian countries.

Liquid and tissue biopsies are used to detect genetic changes in tumors. These two biopsy methods have their own advantages and disadvantages. Compared to tissue biopsy, liquid biopsy provides a noninvasive and easily accessible modality for detecting therapeutic targets and resistance mechanisms [11]. A study reported that liquid biopsy could provide a valuable tool for T790M mutation detection. However, non-plasma samples offer a superior T790M mutation detection rate [12]. Besides, histologic transformation is also an important mechanism of resistance to EGFR-TKIs and cannot be detected by blood-based testing alone. Therefore, tissue re-biopsies remain important for analyzing the resistance mechanisms of lung cancer. [13] As a result, these two methods are complementary to each other.

Interventional pulmonology (IP) is a specialty in which minimally invasive endoscopic and percutaneous procedures are used for the diagnosis and treatment of neoplastic and non-neoplastic diseases of the pulmonary system [14]. Kirita et al. showed that bronchoscopic re-biopsy is feasible for NSCLC and provides sufficient tissue to identify acquired resistance mechanisms [15]. Goag et al. suggested that bronchoscopy is more useful than other re-biopsy approaches [16]. Izumo et al. demonstrated that endobronchial ultrasound (EBUS)-guided transbronchial needle aspiration (EBUS-TBNA) and EBUS with a guide sheath (EBUS-GS) are useful and safe sampling procedures for mutation analysis [17]. Sono-guided percutaneous core needle biopsy (PTCNB), with an 18 G biopsy needle, of peripheral lung lesions shows high diagnostic efficacy [18]. Thoracoscopic biopsy under local anesthesia can help detect the T790M mutation and is associated with limited morbidity [19,20]. Bronchoscopy with EBUS-GS or EBUS-TBNA, sono-guided PNB, or medical pleuroscopy, commonly used by interventional pulmonologists, enables the collection of sufficient tissue samples to permit histopathological and molecular assessment.

In addition to IP methods, video-assisted thoracic surgery biopsy (VATS) and computed tomography (CT)-guided biopsy are commonly used to obtain lung cancer tissue. However, compared to VATS and CT-guided biopsy, the IP methods are less invasive, simpler, and safer. Our previous study showed that EBUS-guided biopsy of peripheral lung lesions led to a significant reduction in iatrogenic pneumothorax compared to CT-guided biopsy [21]. The aims of this study were as follows: (a) to investigate the efficacy of IP procedures as first-line methods for detecting the T790M mutation; (b) to compare differences in the rate of detection of the T790M mutation between IP and non-IP methods; and (c) to evaluate the clinical value of repeat re-biopsy based on the experience of a single center.

## 2. Materials and Methods

### 2.1. Study Design and Patient Enrollment

Between January 2020 and August 2022, we retrospectively evaluated patients with advanced or metastatic EGFR mutation-positive NSCLC who underwent re-biopsy of suspected recurrent or progressive lesions after first- or second-generation EGFR-TKI treatment at the China Medical University Hospital, a tertiary referral center in Taiwan. We investigated the rate of detection of T790M at first-time re-biopsy using IP methods. We also performed repeat re-biopsy of T790M-negative tumors to evaluate the clinical value of repeat re-biopsy. Patients were excluded if re-biopsy was performed for the purpose of clinical trial enrollment without evidence of disease progression or if they had insufficient data for analysis. The study protocol was approved by the institutional ethics committee of the relevant institution (IRB number: CMUH110-REC1–244), and informed consent was waived due to the observational and retrospective study design. The study was conducted in accordance with the Declaration of Helsinki and followed the Strengthening the Reporting of Observational Studies in Epidemiology (STROBE) guidelines.

Data on baseline characteristics of each patient, including age, sex, smoking status, Eastern Cooperative Oncology Group Performance Status (ECOG PS), EGFR mutation subtype, tumor–node–metastasis (TNM) stage at initial diagnosis, first-line EGFR-TKI treatment, re-biopsy methods, re-biopsy sites, T790M status after re-biopsy, time interval between diagnosis and re-biopsy, and histology results, were reviewed in the electronic chart records.

### 2.2. Re-Biopsy Methods and EGFR T790M Mutation Testing

Patients were diagnosed with advanced or metastatic EGFR-positive NSCLC at the initial biopsy, i.e., biopsy at the time of diagnosis. Re-biopsy was defined as a subsequently performed biopsy to screen for the acquired resistance mechanism (the T790M mutation) after the failure of any line of anti-cancer treatment. The first re-biopsy was defined as the biopsy performed for the first time after the failure of the first-line EGFR-TKI treatment, the second re-biopsy as the re-biopsy performed after the first re-biopsy, and so on. The biopsy procedures included IP methods (Figure 1)—bronchoscopy with radial probe-EBUS (R-EBUS), EBUS-TBNA, sono-guided biopsy, and medical pleuroscopy—as well as non-IP methods—CT-guided biopsy and surgery.

The biopsy samples were divided into two groups: tissue sampled by the IP methods; and tissue sampled by the non-IP methods. The success rate of tissue acquisition was the number of patients with positive acquisition divided by the total number of patients who underwent re-biopsy. The samples were examined to determine whether they were malignant and whether enough tumor cells were collected for extraction and EGFR mutation testing. The EGFR T790M mutation status of tumor tissue samples was assessed using cobas^®^ EGFR Mutation Test v2, a PCR-based commercial EGFR mutation detection kit.

### 2.3. Statistical Analysis

All statistical analyses were performed using MedCalc for Windows version 18.10 (MedCalc Software, Ostend, Belgium). Data for normally distributed variables are expressed as the mean ± standard deviation, and data for non-normally distributed variables as the median and interquartile range. Continuous data with a normal distribution were analyzed using the *t*-test. Categorical variables are presented as the number and percentage and were analyzed using the chi-square test. A *p*-value of <0.05 was set as the critical value for statistical significance. Data were visualized with Microsoft Excel for iMac, version 16.30.

## 3. Results

### 3.1. Patient’s Enrollment and Baseline Characteristics

One hundred and thirty-nine advanced or metastatic EGFR mutation-positive NSCLC patients with disease progression after first- or second-generation EGFR-TKI therapy were enrolled in this study; all patients underwent first-time re-biopsy by IP methods to identify the T790M mutation (Figure 2). The median age of the patients was 64.2 years, and the proportion of male patients was 30.9% (Table 1). The EGFR gene mutations in the patients included the exon 19 deletion (48.2%), exon 21 L858R (46.8%), and others (5.1%). The most commonly used EGFR-TKIs were afatinib (38.8%) and erlotinib (34.5%). The median time elapsed between initial diagnosis and re-biopsy was 18.6 months.

On first re-biopsy, 35 (36.0%) patients had the T790M mutation and 6 patients showed histologic transformation. Of the 62 patients whose tumors were shown to be without the T790M mutation on the first re-biopsy, 30 (30.6%) underwent a second re-biopsy. The T790M mutation was detected in eight (36.4%) patients, and histologic transformation was found in four patients on the second re-biopsy, indicating that repeat re-biopsy can increase the rate of detection of the T790M mutation. Six patients underwent a third re-biopsy, and none of the patients had a re-biopsy more than four times. One patient still showed histologic transformation on the third re-biopsy. The tissue acquisition success rate of the first, second, and third re-biopsy was 81.2%, 93.6%, and 100%, respectively. A total of 22 patients received liquid biopsy testing after previous line treatment failure. Among them, six patients (27.2%) had T790M mutation in liquid samples.

### 3.2. The Biopsy Tissue Assessment and Diagnosis by Different Biopsy Methods

All tissue samples from the first re-biopsies were acquired using IP methods because they were convenient and available to the interventional pulmonologist at our institution (Figure 3). The most common methods were R-EBUS (*n* = 88, 63.3%), EBUS-TBNA (*n* = 22, 15.8%), medical pleuroscopy (*n* = 16, 11.5%), and sono-guided biopsy (*n* = 13, 9.4%). The most common site re-biopsied using IP methods was primary lung lesions, whereas metastatic sites were the most common re-biopsy sites when non-IP methods were applied (*p* = 0.009). In the non-IP methods group, most of the metastatic sites re-biopsied were in the brain, spine, and bone. A total of 64 (36.9%) and 25 (14.4%) cytology and cell blocks were obtained from re-biopsy procedures with a diagnostic yield of 75% and 72%, respectively (Table 2). In addition, non-IP methods were used in 43.3% of the second re-biopsies and 83.3% of the third re-biopsies because they were more reliable for ensuring tissue sampling in later-line settings. The success rate of tissue acquisition was 87.6% for IP methods and 100% for non-IP methods (*p* = 0.051). CT-guided biopsy and surgery showed the highest tissue acquisition rates (100% in both cases). Pleuroscopy had a higher success rate (94.4%) in tissue acquisition than other IP methods (Figure 4A,B).

### 3.3. The Biopsy Tissue Molecular Analysis by Different Biopsy Methods

Analysis of the EGFR mutation was feasible in 126 (85.7%) of the 147 samples from the first to the third biopsy, of which 43 (34.1%) harbored the T790M mutation. The T790M mutation positivity of the IP and non-IP methods was 33.6% and 37.5% (*p* = 0.762), respectively (Figure 5A). T790M detection by pleuroscopy yielded the highest positivity rate (46.2%, *n* = 6/13), followed by CT-guided biopsy (40.0%) and surgery (36.4%) (Figure 5B). The rate of detection of the T790M mutation was higher for metastatic sites, such as the liver, pleura, and bone or spine (Figure 5C). The T790M mutation detection rate in comparison between liquid and tissue biopsy from the current study was not significantly different among the three methods (IP methods vs. non-IP methods vs. liquid biopsy: 33.6% vs. 37.5% vs. 27.3%; *p* = 0.782). Of the 147 biopsy samples, 136 (92.5%) were diagnosed with adenocarcinoma and 11 (7.5%) were diagnosed with a different histologic type than that determined at initial diagnosis, including small cell (4.1%, 6/147), neuroendocrine (2.7%, 4/147), and squamous cell (0.6%, 1/147) histologic transformation (Figure 6).

## 4. Discussion

Several previous studies have shown that the diagnostic yields of re-biopsy with VATS and CT-guided fine needle aspiration or core needle biopsy are excellent [22,23]. However, these methods are associated with a high risk of complications, including bleeding, pneumothorax, and morbidity. In contrast, complications are relatively few for IP methods. To the best of our knowledge, this is the first study to investigate the efficacy of all IP procedures as first-line methods for detecting the T790M mutation. We demonstrated that first-line use of IP methods are feasible and provide sufficient tissue for analysis of the acquired resistance mechanism. IP methods and non-IP methods were similar in their detection of the T790Mmutation. Repeat re-biopsy increased the rate of T790M positivity in NSCLC patients after EGFR-TKI failure.

In the present study, all patients underwent first-time tissue re-biopsy with IP methods. Among the re-biopsy methods, bronchoscopy with R-EBUS and EBUS-TBNA were the two most used tools, yielding successful pathological diagnoses in 78–87.5% of the cases. The diagnostic rate was similar to that (73.4%) in a previous study [16]. Another study had an overall detection rate of 87% for re-biopsy of malignant cells by transbronchial biopsy (TBB) and EBUS-TBNA [15]. In Kim et al.’s study, the sensitivity, negative predictive value (NPV), and accuracy of EBUS-TBNA for re-biopsy were 95.6%, 82.7%, and 96.3%, respectively, and there were no major complications during the procedure [24]. Based on these results, bronchoscopy with R-EBUS and EBUS-TBNA have high diagnostic value with few complications.

Previous studies reported an adequate specimen acquisition rate of 80–90% for CT-guided needle biopsy, but 14–20% of patients experienced biopsy-related complications [25,26]. The present study also found that CT-guided needle aspiration and surgery have higher diagnostic yields; however, the T790M mutation detection rate in all diagnostic samples was similar between bronchoscopy with R-EBUS or EBUS-TBNA and CT-guided needle biopsy or surgical resection (32.8–33.3% vs. 36.4–40.0%; *p* = 0.920). A meta-analysis reported that PTCNB can obtain an adequate sample rate of 86.9% for molecular analysis with a T790M mutation detection rate of 46.0% [27]. Goag et al. reported a 43.9% T790M mutation detection rate for bronchoscopy biopsy samples [16]. On the other hand, Kirita et al. obtained a higher rate of detection of the T790M mutation (56%) with TBB and EBUS-TBNA [15]. The inconsistency in the rates of detection of the T790M mutation between these studies may be explained by different testing methods, different biopsy sites, and tumor heterogeneity. However, these results suggest that bronchoscopy with R-EBUS and EBUS-TBNA are useful and provide sufficient tissue to identify resistance mutations.

In the present study, the rate of detection of the T790M mutation in re-biopsy tissue by medical pleuroscopy was 46.2%. To the best of our knowledge, this is the first report of the use of medical pleuroscopy to detect the T790M mutation. Masatsugu et al. suggested that thoracoscopic biopsy can help to detect the T790M mutation with low morbidity [19]. Tissue sampling by thoracoscopy under local anesthesia can reveal the T790M mutation [20]. Thoracentesis with cytology evaluation of malignant pleural fluid is commonly performed to detect the T790M mutation, with a detection rate of 44.4–54.7% [28,29,30]. However, acquired resistance to EGFR-TKIs varies. In addition to the T790M mutation of the target gene, other genomic mutations, alternative pathway activation, or histological transformation should be considered [31]. In the present study, histological transformation was diagnosed in 7.5% of re-biopsy samples. In a large retrospective cohort study, histological transformation occurred in approximately 3% of EGFR-mutated patients who progressed to EGFR-TKI therapy [32]. Therefore, medical pleuroscopy may be advantageous for additional analysis of the resistance mechanism and evaluation of histological transformation if the tissue sample volume is sufficient.

The clinical value of repeat re-biopsy after the first re-biopsy of T790M-negative patients has been previously discussed. The CS-Lung-003 study reported that repeat re-biopsy increased the T790M positivity rate from 43% to 57% [30]. A real-world study also indicated that repeat re-biopsy could increase the detection rate of T790M-positive NSCLC patients from 53.1% to 71% [33]. Repeat re-biopsy may increase the T790M positivity rate even up to 80% [34]. Lee et al. found that the highest success rate of repeat biopsy was 78% and the rate of detection of the T790M mutation in multiple or delayed repeat biopsy samples was 40% [35]. The present study also shows that repeat re-biopsy can increase the number of patients detected with the T790M mutation after second-line treatment. This implies that multiple repeat re-biopsies are needed in some cases because of tumor heterogeneity.

Kim et al. found a higher T790M mutation detection rate with re-biopsy of metastatic lesions than with biopsy of the primary tumor [26]. Another previous study also observed a higher T790M mutation detection rate in metastatic mediastinum lymph nodes than in the primary tumor [16]. Consistent with previous reports, the present study found a higher T790M mutation detection rate, although not statically significant, in re-biopsies of metastatic sites, including the liver, pleural tissue, and bone. The rate of detection of the T790M mutation in different locations varies due to considerable tumor heterogeneity. Therefore, compared to CT-guided biopsy and VAST, the IP procedures in the present study—bronchoscopy with R-EBUS or EBUS-TBNA, ultrasound-guided PNB, and medical pleuroscopy—are more convenient and comprehensive methods that provide the interventional pulmonologist with the means to overcome the obstacle of tumor heterogeneity.

The present study has some limitations. First, it employed a retrospective study design with a limited number of patients and focused on re-biopsy tissue obtained by IP methods. Second, there was an imbalance in the number of tissue samples acquired with IP methods and those obtained with non-IP methods; this might have affected the study’s findings that support and confirm the greater usefulness of IP methods than of non-IP methods as first-line approaches. Third, the data were mostly based on histologic evaluation and single-gene EGFR testing using the the cobas^®^ EGFR Mutation Test v2; they were not based on the next-generation sequencing (NGS) and liquid biopsy were not widely used in clinical practice due to its high price. These tests are not covered by health insurance. As a result, we could not identify other mutations and concurrent generic alternation. Finally, we did not assess the prognosis of patients with or without the T790M mutation, which was considered in a previous study [10]. The essential point of our study was to evaluate the feasibility and utility of IP procedures as first-line methods for detecting the T790M mutation, which could increase the number of patients accessing subsequent osimertinib therapy.

## 5. Conclusions

The present study provides information about the feasibility of IP procedures as first-line re-biopsy methods for patients with NSCLC resistant to EGFR-TKIs. We found that the T790M detection rate was similar between IP methods and non-IP methods and that repeat re-biopsy could increase the detection rate. We suggest deferring non-IP methods until after negative results are obtained with IP methods.

## Figures and Tables

**Figure 1 diagnostics-13-00129-f001:**
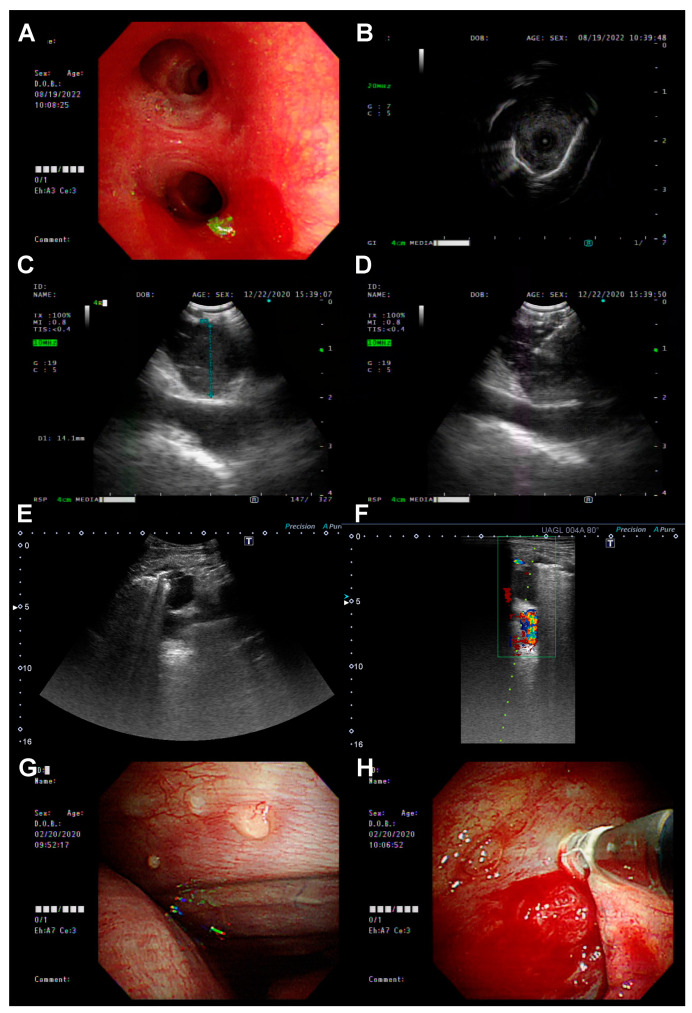
The biopsy procedures included IP methods: (**A**,**B**) Bronchoscopy with radial probe-EBUS (R-EBUS); (**C**,**D**) EBUS-TBNA; (**E**,**F**) sono-guided biopsy; and (**G**,**H**) medical pleuroscopy.

**Figure 2 diagnostics-13-00129-f002:**
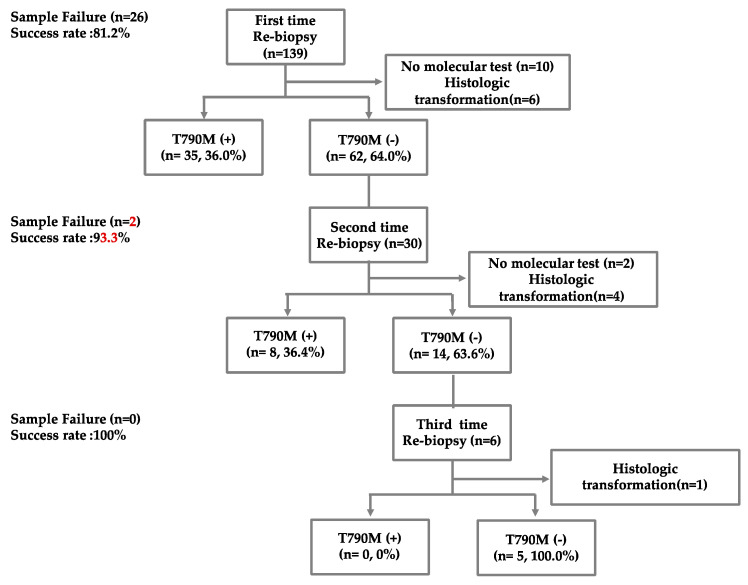
Flowchart showing the rate of detection of the T790M mutation in EGFR mutation-positive non-small-cell lung cancer patients at repeat re-biopsies.

**Figure 3 diagnostics-13-00129-f003:**
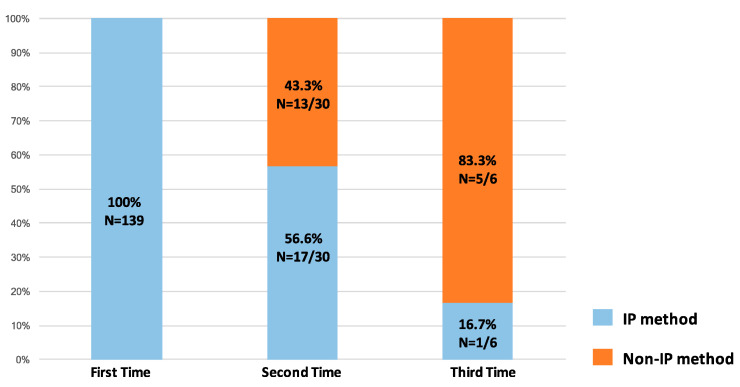
The proportion of use of interventional pulmonology (IP) and non-IP re-biopsy methods in first, second, and third biopsies.

**Figure 4 diagnostics-13-00129-f004:**
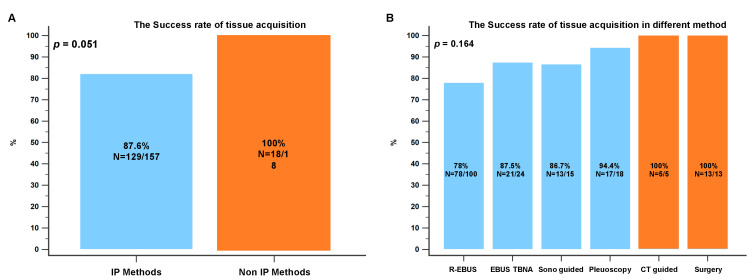
(**A**) The tissue acquisition success rates of all IP and all non-IP methods. (**B**) The tissue acquisition success rates of different IP and non-IP biopsy methods. CT: computed tomography; IP: interventional pulmonology; R-EBUS: radial probe endobronchial ultrasound; TBNA: transbronchial needle aspiration.

**Figure 5 diagnostics-13-00129-f005:**
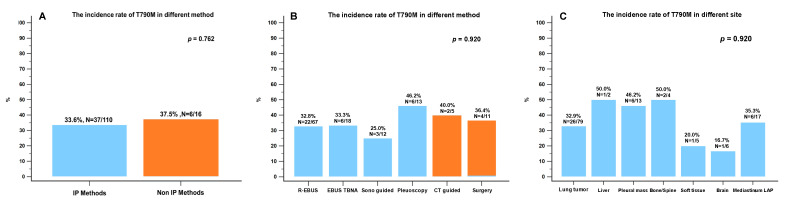
(**A**) The rate of incidence of the T790M mutation detected by all IP and all non-IP methods. (**B**) The rate of incidence of the T790M mutation detected by different IP and non-IP biopsy methods. (**C**) The rate of incidence of the T790M mutation rate in different biopsy sites. CT: computed tomography; IP: interventional pulmonology; LAP: lymphadenopathy; R-EBUS: radial probe endobronchial ultrasound; TBNA: transbronchial needle aspiration.

**Figure 6 diagnostics-13-00129-f006:**
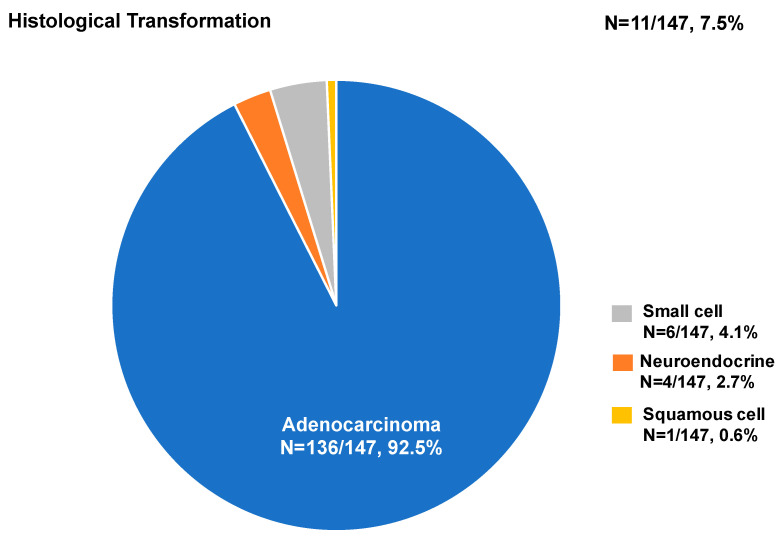
The histologic diagnosis of tissue acquired in biopsies.

**Table 1 diagnostics-13-00129-t001:** Characteristics of patients enrolled in this study.

	All (*n* = 139)
Age, year (SD)	64.2 (10.4)
Male (%)	43 (30.9)
Smoking (%)	29 (20.9)
ECOG PS ≥ 2 (%)	7 (5.0)
*EGFR* mutation (%)	
Del 19	67 (48.2)
L858R	65 (46.8)
Other	7 (5.1)
Stage	
IIIA	3(2.2)
IIIB	8 (5.8)
IIIC	2 (1.4)
IVA	55 (39.6)
IVB	61 (43.9)
Post-OP recurrence	10 (7.2)
EGFR-TKI	
Gefitinib	34 (24.5)
Erlotinib	48 (34.5)
Afatinib	54 (38.8)
Other	3 (2.2)
Time to re-biopsy, month (95% CI)	18.6 (16.5–20.9)

CI: confidence interval; ECOG PS: Eastern Cooperative Oncology Group Performance Status Scale; EGFR: epidermal growth factor receptor; OP: operation; SD: standard deviation; TKI: tyrosine kinase inhibitor.

**Table 2 diagnostics-13-00129-t002:** The differences in re-biopsy sample characteristics between IP methods and non-IP methods.

	IP Methods (*n* = 157)	Non-IP Methods (*n* = 18)	*p*-Value
Biopsy method			<0.001
Bronchoscopy	100 (57.1)	0 (0)	
EBUS-TBNA	24 (13.7)	0 (0)	
Ultrasound-guided	15 (8.6)	0 (0)	
Medical pleuroscopy	18 (10.3)	0 (0)	
CT-guided biopsy	0 (0)	5 (2.9)	
Surgery	0 (0)	13 (7.4)	
Time to re-biopsy, months	18.1 (11.8–30.1)	21.6 (16.0–20.4)	0.601
Biopsy site			0.009
Primary site	109 (69.4)	7 (38.9)	
Metastatic site	48 (30.6)	11 (61.1)	
Biopsy organ			<0.001
Lung mass	108 (93.9)	7 (6.1)	
Liver	4 (100)	0 (0)	
Pleural mass	18 (100)	0 (0)	
Spine/bone	0 (0)	4 (100)	
Soft tissue	6 (100)	0 (0)	
Brain tumor	0 (0)	7 (100)	
Mediastinum lymph node	21 (100)	0 (0)	
Histology result			0.099
Adenocarcinoma	121 (77.1)	15 (83)	
Neuroendocrine	3 (1.9)	1 (5.6)	
Small-cell carcinoma	4 (2.5)	2 (11.1)	
Squamous cell carcinoma	1 (0.6)	0 (0)	
No malignancy	28 (17.8)	0 (0)	

CT: computed tomography; EBUS-TBNA: endobronchial ultrasound-guided transbronchial needle aspiration; IP: interventional pulmonology.

## Data Availability

Not applicable.

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
