# Peer review of "The Feasibility of Interventional Pulmonology Methods for Detecting the T790M Mutation after the First or Second-Generation EGFR-TKI Resistance of Non-Small Cell Lung Cancer"

_diagnostics, 2022, doi:10.3390/diagnostics13010129_

Round 1

Reviewer 1 Report

The authors showed here their results in determining EGFR T790M mutation in second or third rebiopsies after progression on EGFR TKIs.

There are 2 major criticisms:

1. the advent of osimertinib in first line treatment of patients with EGFR mutations basically prevent the necessity to re-test T790M mutation

2. re-biopsy in EGFR mutated patients should include several findings, such as histologic change, c-met amplification, and other EGFR mutations or concurrent genetic alterations in other genes tested with NGS or multiplex technologies

At the end, while the % of re-biopsy and the % of cases with adequate material for molecular analsys is intersting, the detection of T790M mutation is not necessary with the advent of osimertinib and results of little interesting fo the readers.

So, I suggest the authors to re-organize the study evidencing the % of re-biopsy and % of tissue adequacy, also including how many cytology vs biopsy specimens were obtained at rebiopsy and how many cell-block were performed, how many histologic changes have been found.

Reviewer 2 Report

I thank the authors for submitting their manuscript to review.

The manuscript is quite well written, from a syntactic point of view, but it seems a bit confusing to me. Anyway, I would suggest a minor spelling English revision.

Nevertheless, it is not clear to me which is the interest for readers. Currently, most of the research and the clinic are going towards developing less invasive techniques to reduce the discomfort for the patient as much as possible, in this regard, liquid biopsy is increasing its importance in clinical practice, for diagnostic purposes, in the disease follow-up and for assessing drug resistance, particularly for TKIs (please read the following: PMID 36277960; 33486418; PMID ).

Lines 145, 215, 221, 259, 260, 263, 265, 270, 295: the authors repeat numerous times the importance of liquid biopsy in increasing the positivity T790M mutation rate. My question is: is it reasonable to repeat tissue biopsies, even “less” invasive to improve the mutational rate discovery?

Lines 51-54, please provide the Reference for this statement.

The authors state that histologic transformation is an important characteristic in resistance to TKIs., and thus the tissue biopsy is crucial for this aim. Most of the current studies are focused on the increasing importance of liquid biopsy, thus, I am wondering to ask: why they report “against” this trend?

Moreover, the authors point the light on the importance of histologic transformation, but it is not clear to me the topics they are investigating: the mutational rate or the histologic transformation.

Lines 188-190: the authors state that the T790M mutation rates between IP and non-IP methods are very similar (33.6% vs 37.5%, non-statistically different), what about the comparison of IP vs liquid biopsy?

Moreover, the authors never mention other mutations currently under investigation, that might be involved in TKIs’ resistance. I would suggest the following papers: PMID 36387265; PMID 36437214; PMID 36419902.

Round 2

Reviewer 1 Report

None

Reviewer 2 Report

I thank the authors for the extensive revision of the manuscript, they deeply revised all the sections., following the reviewers' suggestions.

The topic is interesting, and the authors provided several pieces of evidence on the theme.

Anyway, in my opinion, the present manuscript does not provide a significant scientific interest for the readers.